# Impact of Ad Libitum Hydration on Muscle and Liver Damage and Electrolyte Balance in Ultra-Trail Events: A Heatmap Analysis of Biomarkers and Event Characteristics—A Pilot Study

**DOI:** 10.3390/biology14020136

**Published:** 2025-01-28

**Authors:** Alejandro García-Giménez, Francisco Pradas, Miguel Lecina, Nicolae Ochiana, Carlos Castellar-Otín

**Affiliations:** 1ENFYRED Research Group, Faculty of Health and Sports, University of Zaragoza, 22002 Huesca, Spain; alejandro.garcia@unizar.es (A.G.-G.); franprad@unizar.es (F.P.); castella@unizar.es (C.C.-O.); 2Department of Physical Education and Sports Performance, Faculty of Movement, Sports and Health Sciences, “Vasile Alecsandri” University of Bacau, 600115 Bacau, Romania; sochiana@ub.ro

**Keywords:** hyponatremia, fluid imbalance, ultra-endurance, body weight loss, hydration strategies, exertional rhabdomyolysis

## Abstract

Hydration strategies have become a significant concern in ultra-endurance sports over the past decade due to medical conditions, such as dehydration (DH), exercise-associated hyponatremia (EAH), exertional rhabdomyolysis (ER), and liver damage. Identifying which race characteristics contribute most to these issues remains challenging due to the great diversity of these races and the absence of a standard classification. This study employs heatmap analysis to investigate the relationship between changes in various biomarkers and several factors, like altitude, duration, and cumulative elevation. The results offer valuable insights for runners, coaches, and medical staff to better prepare for extreme races and prevent complications, such as DH, EAH, ER, and liver damage.

## 1. Introduction

Ultra-trail events (UT) present unique and significant challenges to maintaining hydration and electrolyte balance because of their extreme duration (>42.195 km or at least 6 h) [1,2] UT athletes are exposed to environmental extremes [3], including heat, humidity, and altitude, which increase sweat rates and the potential occurrence of dehydration (DH), electrolyte imbalances, and even medical conditions, such as heat stroke or death [4]. The risk of DH is ever-present due to the prolonged nature of these events, where fluid losses can accumulate over time [5], and has been a matter of concern for prestigious institutions, such as ACSM [6] or ISSN [7]. However, the risk of overhydration (OH) is equally concerning, especially when athletes consume large amounts of fluids or start the event overhydrated [8]. The complexity of these events demands a specific understanding of how to balance fluid and electrolyte intake to prevent runners from suffering DH or OH. OH may provoke a medical condition called exercise-associated hyponatremia (EAH) that occurs when an athlete’s sodium [Na+] levels drop to dangerously low levels during or after prolonged exercise. EAH is defined by serum [Na+] below 135 mmol·L^−1^, with severe cases occurring when levels drop below 120 mmol·L^−1^, with normal levels ranging within 135–145 mmol·L^−1^ [9].

EAH cases have been reported in UT events ranging from 0.0% [10] to 20% [11]. The most common diagnostic tool to measure fluid balance is the percentage of body weight lost (BWL) during exercise. Mild DH is defined by a 1–2% BWL, moderate DH by a 3–5% loss, and severe DH by losses exceeding 5% [12]. The consequences of DH are particularly pronounced in endurance sports, where athletes engage in prolonged physical activity. Studies have shown that even mild DH can impair performance by reducing maximal oxygen consumption (VO_2max_), increasing perceived exertion, and impairing thermoregulation. In extreme cases, DH can lead to a dangerous health condition known as heat stroke, which requires immediate medical intervention [13].

Another diagnostic tool to assess DH is urine specific gravity (U_sg_). U_sg_ is a widely used marker to evaluate hydration status, with a threshold value of 1.020 g·mL^−1^ often employed to distinguish between euhydration and DH [14]. Values above this threshold indicate DH, as they reflect higher concentrations of solutes, suggesting insufficient fluid intake relative to losses. Studies have demonstrated the utility of U_sg_ in various athletic contexts, including endurance sports, where it provides a reliable, non-invasive measure of hydration status both before and after exercise [15]. The primary cause of EAH is excessive consumption of hypotonic fluids, such as water or sports drinks with a low [Na+] content, in amounts that exceed the renal excretion capacity. This excessive fluid intake dilutes blood’s [Na+], leading to EAH. Conversely, athletes experiencing BWL greater than 2% may develop hypernatremia, a condition characterized by serum [Na+] exceeding 145 mmol·L^−1^ [16].

Electrolyte disorder affects performance and may affect the muscle structure, debilitating the sarcolemma and producing the leakage of muscle cell contents, myoglobin, sarcoplasmic proteins—such as creatine kinase (CK), lactate dehydrogenase (LDH), aldolase, alanine, and aspartate aminotransferase—and electrolytes—such as calcium (Ca)—which during the acute phase shifts into damaged muscle cells and precipitates with phosphates in necrotic tissues [17]. Hypocalcemia often develops as Ca shifts into damaged muscle cells during the acute phase of exertional rhabdomyolysis (ER). It precipitates with phosphates in necrotic tissues, contributing to the disruption of normal cellular function. ER is a medical condition which has been linked to acute kidney [18,19] and liver (aspartate aminotransferase, AST; alanine aminotransferase, ALT) [20] damage in sports. The diagnosis of ER is varied, without consensus on whether the diagnosis is based on clinical presentation, laboratory findings, or a combination of the two. Laboratory diagnosis of ER shows elevations in serum CK, and there is no specific established serum level cut-off [21]. Many physicians use three to five times the upper limit of normal Values of 100 to 400 U/L (approximately 1000 U/L) for diagnosis. EAH and ER cases have been reported in different sports [22,23]. It has been suggested that a low blood [Na+] concentration may contribute to muscle cell swelling, which, combined with repetitive mechanical stress from running, may accelerate skeletal muscle damage [24]

Trail running, particularly in ultra-endurance formats, presents unique and significant challenges to maintaining hydration and electrolyte balance. Traditionally, athletes have been advised to follow regimented hydration schedules, often based on predetermined fluid intake rates. However, recent research suggests that this approach may not be ideal for all athletes, as it does not account for individual variability in sweat rates, fluid needs, and risk factors for EAH [25]. Some studies have advocated for an ad libitum (ADL) hydration strategy [26] approach to hydration, where athletes drink according to thirst. This strategy is believed to balance fluid intake with the body’s actual needs more naturally, potentially reducing the risk of DH and EAH. On the other hand, other studies [27] have recommended using planned hydration (PH), where fluid intake is based on pre-event calculations.

Given the dual risks of DH and EAH in UT, this study investigates whether an ADL can effectively mitigate these risks in highly trained athletes during a multi-stage UT event. The study will assess whether ADL drinking can maintain adequate hydration without leading to the excessive fluid intake associated with EAH measured by BWL, U_sg_, and serum [Na+], the simultaneous presence of ER analyzed by serum CK, LDH, and Ca, and liver damage (AST and ALT). Additionally, this study explores the factors influencing UT performance and their relationships with ER, EAH, and DH, providing valuable insights into these dynamics. The underlying hypothesis posits that an individualized hydration strategy tailored to these factors will help prevent DH, EAH, and ER, thereby enhancing event performance and mitigating the risk of adverse health outcomes. This study investigates and offers new insight into UT and EAH, ER, liver damage, and drinking strategies by exploring the likely relationships between these medical conditions and all the characteristics involved in this extreme outdoor sport. By using a heatmap, we establish the relationships between all the factors that influence the occurrence of EAH, ER, and liver damage.

## 2. Materials and Methods

### 2.1. Experimental Design

This event, carried out in the north of Spain, consisted of completing a total of 635 km with 40,586 m elevation gain and 39,811 m elevation loss in 9 days (daily mean: 70.6 ± 4.96 km; 4509.56 ± 1010.43 m elevation gain; 4423.44 ± 984.86 m elevation loss. The stages’ characteristics are described in Table 1. The research was carried out according to the Declaration of Helsinki, and the project’s approval was obtained from the Ethics Committee of the Department of Health and Consumption of the Government of Aragon (Spain) (protocol code 18/2015).

### 2.2. Participants

A total of four highly trained male athletes (age 38 ± 4.11 years; VO_2max_ 61.17 ± 8.96 mL·kg^−1^·min^−1^; UT experience 5 ± 1.26 years) voluntarily participated in this study. They were free of any pharmacological, medical, or dietary treatment. Participants were adequately informed about the study objective, procedures, and risks. Personal written informed consent was previously provided to participants, and they could withdraw from the study at their own will at any time. Inclusion criteria were as follows: (1) older than 18 years old; (2) previous experience in at least two ultramarathons (longer than 42 km); and (3) not affected by any chronic medical treatment or condition. The complete participants’ characteristics can be found in Table 2.

### 2.3. Anthropometry and BWL

Anthropometric measurements were carried out by a skilled and experienced technician according to the standards for anthropometric assessment [15] 2 h prior to the start of the event. These measurements included height, weight, and skin folds. Height was measured using a wall-mounted stadiometer (Seca 220, Seca, Hamburg, Germany) to the nearest 0.1 cm. Weight was measured to the nearest 0.01 kg on a calibrated electronic digital scale (Seca 769, Hamburg, Germany) nude and barefoot. BWL was calculated by the difference between pre- and post-stage weight to the nearest 0.01 kg. These measurements were carried out 30 min before the beginning and immediately after each stage to the nearest 0.01 kg using a calibrated digital scale (Seca 769, Hamburg, Germany).

### 2.4. Urine Analysis

The first urine sample in the morning (baseline) and the first urine sample after stages 1, 3, 5, 7, and 9 were taken from all the participants. Samples were collected in polyethylene tubes, measured, codified, and frozen at −80 °C until analyzed. The tubes had been previously washed with diluted nitric acid. Before the analyses, samples were thawed and homogenized by shaking. For U_sg_ analysis, a 10 mL urine sample was used to obtain each participant’s values. To achieve this aim, a pre-calibrated refractometer (URC-Ne, Atago, Tokyo, Japan) was employed in situ as previously described [18].

### 2.5. Blood Samples

Baseline samples were taken 90 min before stage 1 and consequent samples were collected immediately after stages 1, 3, 5, 7, and 9 (approximately 10 min post-exercise). An extra sample was extracted 48 h after the ninth stage to monitor recovery.

Two 5 mL venous blood samples were drawn from the antecubital vein of each participant to assess serum white blood cells (WBC), neutrophils, lymphocytes, monocytes (MON), eosinophils (EOS), basophils, erythroblasts, red blood cells (RBC), hemoglobin, hematocrit, mean corpuscular volume, mean corpuscular hemoglobin (MCH), mean corpuscular hemoglobin concentration (MCHC), red cell distribution width, platelets, mean platelet volume, [Na+], CK, LDH, Ca, AST, and ALT. The samples were collected in vacutainer tubes containing ethylenediaminetetraacetic acid (EDTA) as an anticoagulant. Following collection, blood samples were transferred into metal-free polypropylene tubes (pre-washed with diluted nitric acid) and centrifuged at 2500 RPM for 10 min at room temperature to isolate the serum. Samples were left to coagulate for 25–30 min, then allotted into Eppendorf tubes (also pre-washed with diluted nitric acid) and stored at −80°C for later biochemical analysis. Plasma volume (PV) change was calculated using the Dill and Costill equation [28].

### 2.6. Statistical Analysis

Data were processed using Python language (Python Software Foundation, version 3.X) at the cloud-based platform Databricks (Databricks, San Francisco, CA, USA). The normality of variable distribution was assessed using the Shapiro–Wilk test. For variables that did not follow a normal distribution (Shapiro–Wilk test, *p* < 0.05), a permutation test was used to compute the *p*-values, as this non-parametric method does not assume normality and is well-suited for small sample sizes. In cases where the data approximated normality but the small sample size limited the robustness of parametric tests, Monte Carlo simulations were employed (*p* < 0.05) to estimate *p*-values by generating a large number of random samples, enhancing the accuracy of the statistical inference. Effect sizes were calculated using Cohen’s D, where values of 0.2–0.49, 0.5–0.79, and 0.8–1.0 were interpreted as small, medium, and large effects, respectively. Additionally, Pearson correlation coefficients were interpreted as small (r = 0.10–0.29), medium (r = 0.30–0.49), and large (r ≥ 0.50) [29]. Visualizations were created using the matplotlib and seaborn packages. Pearson correlation coefficients were computed to generate the correlation matrix using the corr method in Python. Statistical significance was set at *p* < 0.05.

## 3. Results

### 3.1. White and Red Blood Cell Lineages

WBC counts increased significantly (*p* < 0.05) at all stages compared to the baseline except in stage 9. MON showed significant increases at stage 1 and stage 3, while EOS exhibited a significant suppression at stage 1 and stage 3 (*p* < 0.05) (Table 3).

In the RBC lineage, significant reductions in MCH and MCHC were observed at stages 5 and 7 (*p* < 0.05) (Table 4). These results reflect acute hematological responses to the endurance event, particularly in parameters related to immune activity and erythrocyte morphology.

### 3.2. BWL

A significant difference in BWL was observed between stage 1 and stage 3 (*p* < 0.05; d = −3.32), indicating a substantial reduction in body weight early in the event. No significant differences were found between subsequent stages (3 vs. 5, 5 vs. 7, and 7 vs. 9), suggesting that BWL stabilized as the event progressed but remained at the −2% DH threshold (Figure 1). 

### 3.3. U_sg_

At baseline, most participants had U_sg_ values above this threshold, indicating a DH state. Additionally, U_sg_ increased in the early stages (1 and 3), with medians exceeding 1.020 g·mL^−1^, highlighting dehydration onset during the initial phases of the event. U_sg_ decreased at stage 5, suggesting partial rehydration, but remained elevated near or above the DH threshold through stages 7 and 9, indicating persistent hydration challenges. After 48 h, U_sg_ values dropped below 1.020 g·mL^−1^ for most participants, reflecting a recovery to a hydrated state. Despite fluctuations, no stage comparisons with the baseline reached statistical significance (*p* > 0.05), though the effect size between S0 and S7 (d = −1.61) suggests a notable increase in hydration stress during the event. (Figure 2).

### 3.4. [Na+]

[Na+] analysis reported a 0% incidence of EAH (concentration < 135 mmol·L^−1^), with values remaining within ~136–144 mmol/L at baseline and during the whole event (Figure 3). Statistical analysis shows a trend toward significance between stages 1 and 3 (*p* < 0.05, d = −2.31) and a moderate effect size between stages 7 and 9 (*p* < 0.05, d = 1.34). These patterns suggest a first progression towards hypernatremia and a subsequent recovery phase.

### 3.5. PV

A notable increase in PV is observed between stages 1 and 3 (*p* < 0.05, d = −2.81), indicating an early-stage shift. Mid-stage comparisons (stages 3 to 5, 5 to 7, and 7 to 9) show non-significant changes despite moderate-to-large effect sizes (d = 0.42−1.27), reflecting variability in PV dynamics. A significant decrease occurs from stage 9 to 48 h post-event (*p* < 0.05, d = 2.23). (Figure 4).

### 3.6. Serum CK and LDH

Both CK and LDH showed significant increases throughout the entire event, as evidenced by consistently elevated levels at all stages compared to the baseline (*p* < 0.05) (Figure 4 and Figure 5). CK demonstrated particularly large effect sizes, with the most pronounced changes observed at stage 3 (d = −9.20), and substantial elevations persisting even after 48 h (d = −2.84). (Figure 5). Similarly, LDH levels rose significantly at every stage, with the greatest effect at stage 1 (d = −7.22), and remained elevated at 48 h post-event (d = −2.80). (Figure 6).

### 3.7. Serum Ca

Serum Ca levels significantly increased at the end of stage 1 compared to the baseline (*p* < 0.05, d = −2.29) but remained below the baseline during the intermediate stages, with no significant changes observed in stages 3, 5, or 7 (Figure 6). At the end of stage 9, Ca exhibited a significant decrease compared to the baseline (*p* < 0.05, d = 3.72). (Figure 7).

### 3.8. Serum AST and ALT

Significant increases (*p* < 0.05) in AST (Figure 8) and ALT (Figure 9) levels across the stages compared to the baseline have been observed. For AST, the sharpest increase occurs between the baseline and stage 3 (*p* < 0.05, d = −16.76). For ALT, the highest levels occur at stage 7, with significant differences observed from the baseline to stages 3 (*p* < 0.05, d = −2.61) and 9 (*p* < 0.05, d = −3.00). Both enzymes exhibit large to extremely large effect sizes, with AST ranging from d = −2.39 to −16.76 and ALT ranging from d = −1.67 to −3.00, indicating substantial liver activity. (Figure 9).

### 3.9. Relationship Between the Biomarkers and Extrinsic Event Characteristics

According to Pearson’s correlation analysis (Figure 10), distance was negatively correlated with BWL (r = −0.62), LDH (r = −0.59), and ALT (r = −0.66), while it showed a positive correlation with Ca (r = 0.60). This indicates that as the distance increased, BWL and LDH decreased, while Ca increased. Similarly, elevation gain was positively correlated with BWL (r = 0.52) and [Na+] (r = 0.67). Elevation loss exhibited similar trends, being positively correlated with both BWL (r = 0.52) and [Na+] (r = 0.53).

Furthermore, accumulated elevation gain showed a positive correlation with BWL (r = 0.57), LDH (r = 0.64), and ALT (r = 0.64) but was negatively correlated with Ca (r = −0.62). Accumulated elevation loss also demonstrated positive correlations with BWL (r = 0.55), LDH (r = 0.65), and ALT (r = 0.63), and a negative correlation with Ca (r = −0.62). Temperature was negatively correlated with BWL (r = −0.40) and LDH (r = −0.61) but was positively correlated with Ca (r = 0.41). In the case of velocity, it showed a negative correlation with BWL (r = −0.76), [Na+] (r = −0.54) and AST (r = −0.69), while it was positively correlated with Ca (r = 0.61).

Over subjects’ variables and biomarkers, BWL was negatively correlated with Ca (r = −0.68) and positively correlated with PV (r = 0.61), CK (r = 0.47), AST (r = 0.67), and ALT (r = 0.60). Na was positively correlated with CK (r = 0.43), while Ca exhibited negative correlations with both CK (r = −0.53), LDH (r = −0.40), and AST (r = −0.63). CK was positively correlated with serum LDH (r = 0.50) and AST (r = 0.90). Moreover, LDH was positively correlated with PV (r = 0.56).

## 4. Discussion

### 4.1. EAH

Despite fluctuations in [Na+] during the event—characterized by a progressive increase up to stage 7 followed by a significant decrease from stage 7 to 9 (*p* < 0.05)—no athlete experienced EAH, as [Na+] values remained above 135 mmol·L⁻^1^. Similarly, no cases of hypernatremia ([Na+] > 145 mmol·L⁻^1^) were observed, with all values staying within the normal range according to standard criteria [30].

The incidence of EAH in ultra-endurance sports has been extensively studied, with reported rates varying between 0% and 51% [21]. Arnaoutis et al. [31] examined EAH incidence on a 44 km mountain marathon where 8% of runners (5 out of 62) developed asymptomatic EAH, with a significant drop in serum [Na+]. Lecina et al. [24] conducted a systematic review focusing on UT events and identified a correlation between event distance and EAH incidence, with rates of 2.69% in medium-distance UTs versus 12.19% in long-distance UTs. On the other hand, Hoffman et al. [25] observed an EAH of 15.1% (range 4.6–51.0%) in 161 km ultramarathons, being more common in dehydrated runners and in hotter temperatures. Authors advised against excessive [Na+] supplementation and OH beyond thirst-driven intake.

### 4.2. Hydration Status

BWL is an important marker of fluid loss during physical exertion. The results revealed that BWL initially dropped below 4% in the first stage, indicating significant fluid loss. Although it improved and stabilized (*p* < 0.05) after the third stage, it remained over the 2% threshold during the whole event, suggesting that athletes were still in a dehydrated state. U_sg_ measurements revealed no statistical differences from baseline values, but the athletes’ U_sg_ values remained above 1.020 (the DH threshold) in stages 1, 3, 7, and 9. This suggests that DH persisted at various stages of the event, particularly in the early and late phases. This comprehensive assessment underscores the difficulty of maintaining hydration in ultra-endurance events and supports findings by Rojas-Valverde et al. [19] who also identified significant DH (66.7% incidence) in trail runners completing a distance of 35.3 km and 1815 m of elevation gain when utilizing multiple hydration markers.

Our results are consistent with similar studies in UT running, such as the one reported by Martínez-Navarro et al. [32]. They observed that 38.2% and 47.8% of participants finished dehydrated in a 118 km (5439 m elevation gain) UT race when assessing this issue with BWL and U_sg_, respectively. In addition, they found that BWL was inversely correlated with finishing time (r = −0.34, *p* ≤ 0.05). Interestingly, BWL has been reported to be a key performance factor in some UT distances (50 km) but not in other distances (80 km) [27]. The authors reported that while 80 km racers also experienced BWL, their hematocrit levels decreased, suggesting plasma volume expansion or hemolysis. Non-finishers showed a trend toward higher hematocrit levels, possibly indicating severe DH, though this was not confirmed by other DH markers. Despite these findings, a moderate (2–3) BWL appears to possibly benefit performance.

### 4.3. Elevation Gain/Loss and Muscle and Liver Damage: Key Factors in Ultra-Endurance Events

Elevation gain and loss are critical factors in UT events, where significant vertical displacement imposes unique stress on the body. Elevation gain and loss were positively correlated with BWL (r = 0.52) and [Na+] (r = 0.67 and r = 0.53, respectively), indicating that they increase fluid loss and worsen electrolyte imbalance. Additionally, both accumulated elevation gain and loss were positively correlated to LDH (r = 0.64; r = 0.65), AST (r = 0.64; r = 0.65) and ALT (r = 0.64; r = 0.63), reflecting the strain placed on the muscles and liver during both ascent and descent [33,34,35].

Furthermore, serum CK, LDH, AST, and ALT levels increased significantly over the course of the event and remained elevated 48 h post-event (*p* < 0.05), indicating that muscle and liver damage persisted long after the event’s conclusion. These findings support the idea that ultra-endurance events, particularly those with significant elevation changes, contribute to substantial muscle/liver damage [36]. The sustained elevation of LDH throughout the event is consistent with previous studies showing that LDH is a key marker of muscle injury during prolonged physical activity. Likewise, CK is another well-established marker of muscle damage, and its increase throughout the event further emphasizes the severity of the muscle injury experienced during ultra-endurance events [37,38]. The moderate positive correlation between CK and LDH elevations (r = 0.50) further highlights their interconnected roles as markers of muscle damage, as both enzymes are released into circulation following sarcolemma disruption during intense physical exertion [39,40,41]. The strong correlation between CK and AST (r = 0.90) confirms the interplay reflected in the previous literature linking muscle and liver activity [20,42,43].

Moreover, Ca levels are negatively correlated with accumulated elevation gain and loss (r = −0.62), suggesting that significant elevation changes may reduce serum Ca. Also, its levels significantly rose at the end of stage 1 (*p* < 0.05) but subsequently remained below the baseline values in the mid-stages and dropped significantly by stage 9 (*p* < 0.05). This aligns with the ER findings, where acute hypocalcemia occurs due to Ca influx into damaged muscles and phosphate precipitation. Given Ca’s role in muscle function, its depletion during prolonged exertion may exacerbate fatigue and injury risk, highlighting its importance in assessing recovery in UT events [44].

### 4.4. Linking Hydration Status and Muscle and Liver Damage: The Interplay Between EAH and ER

The results of this study suggest a complex interplay between hydration status and muscle damage during ultra-endurance events, highlighting the importance of both hydration management and muscle recovery strategies. As fluid loss (reflected in BWL and U_sg_) increased throughout the event, the risk of EAH, assessed by serum [Na+], remained in the ‘safe zone’.

Simultaneously, muscle (CK and LDH) and liver (AST and ALT) damage markers, which remained elevated throughout the event, underscore the contribution of sustained mechanical muscle stress, including significant elevation changes, to ER [34]. The correlations between these biomarkers and elevation gain/loss suggest that vertical displacement exacerbates muscle injury, thus increasing the risk of ER and liver damage. Furthermore, the correlation between BWL, Ca (r = −0.68), AST (r = 0.67), and ALT (r = 0.60) further highlights the interplay between DH and this damage [45,46]. Significant fluid loss likely exacerbates Ca imbalances by reducing plasma volume and altering ion transport, contributing to the observed hypocalcemia. This relationship suggests that DH not only impairs overall physiological performance but may also heighten the risk of muscle and liver damage by disrupting Ca, AST, and ALT homeostasis, emphasizing the critical role of hydration strategies in mitigating exercise-related fatigue and injury during ultra-endurance events [34,47].

These findings reinforce the bidirectional relationship between EAH and ER in ultra-endurance events, where disruptions in hydration can contribute to muscle and liver damage. Both conditions appear to be aggravated by various factors, such as elevation gain/loss and prolonged physical exertion, emphasizing the need for comprehensive strategies to manage both hydration and muscle recovery in these extreme events. According to these results, runners and coaches must assess all the characteristics of these races and establish hydration strategies which carefully consider all the factors involved in every UT.

## 5. Conclusions

This study highlights the intricate relationship between hydration status, muscle damage, and performance in UT events. ADL hydration effectively prevented EAH, as [Na+] remained within safe limits throughout the event. However, DH, as indicated by BWL > 2% and elevated U_sg_ > 1.020 g·mL^−1^, persisted during several stages, particularly early in the event. Muscle damage markers, including CK, LDH, and Ca, showed significant changes across all stages. CK, LDH, AST, and ALT increased substantially, while Ca levels fluctuated, dropping below the baseline during later stages, indicating sustained tissue, liver, and electrolyte stress.

The correlations revealed that BWL exacerbated muscle and liver damage and hypocalcemia, while elevation changes intensified fluid loss and muscle and liver injury. The biological mechanisms that can explain the onset of these medical conditions include a lack of available energy due to the long duration of races, as well as imbalances in electrolytes and fluids. Additionally, changes in the sarcolemma can lead to the release of myocyte contents into the bloodstream. Concerning liver damage, further investigation is needed to understand the physiological processes that alter liver enzymes, as this damage can persist for days following a race.

These findings underscore the importance of combining ADL hydration with tailored recovery strategies to mitigate DH and muscle/liver damage. Both are critical for maintaining performance and reducing health risks in ultra-endurance events. Comprehensive approaches addressing individual fluid balance and recovery needs are essential to optimizing outcomes in this demanding sport.

## 6. Limitations

This study is subject to several limitations. First, the sample size was small, which may limit the generalizability of the findings. Second, the high level of physical fitness of the participants could have influenced the results, as their experience in ultra-trail events may affect their physiological responses. Additionally, the absence of a control group with a programmed drinking regimen limits our ability to compare the effects of hydration strategies. These factors should be considered when interpreting the results, and future studies could address these limitations by including larger, more varied samples and control groups.

## Figures and Tables

**Figure 1 biology-14-00136-f001:**
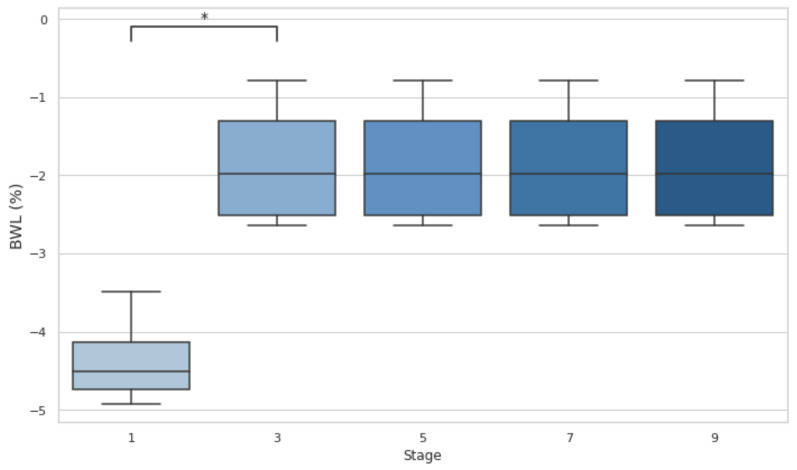
Body weight losses after stages 1, 3, 5, 7, and 9. Dehydration threshold settled at <−2%. BWL: body weight loss; * *p* < 0.05.

**Figure 2 biology-14-00136-f002:**
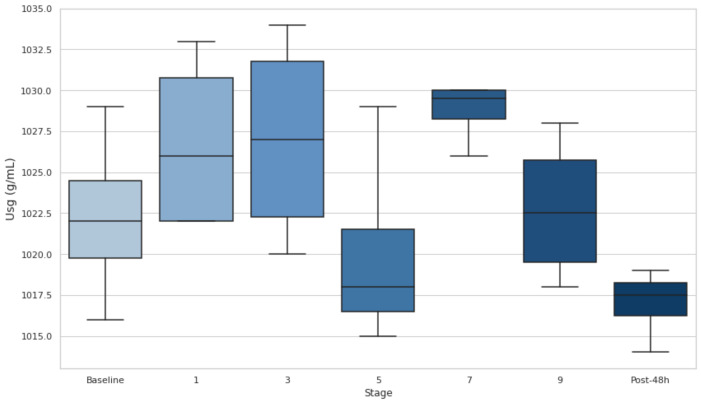
Urine specific gravity at baseline, after stages 1, 3, 5, 7, and 9, and 48 h post-event. Dehydration threshold settled at U_sg_ > 1.020 g·mL^−1^. U_sg_: urine specific gravity.

**Figure 3 biology-14-00136-f003:**
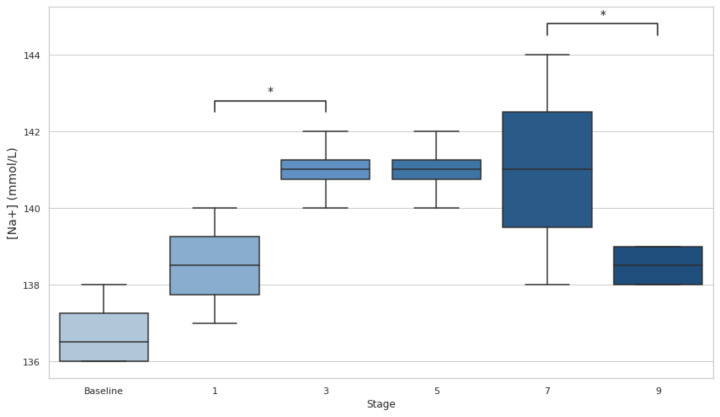
Serum sodium at baseline, after stages 1, 3, 5, 7, and 9, and 48 h post-event. Hyponatremia threshold settled at <135 mmol·L^−1^. [Na+]: serum sodium; * *p* < 0.05.

**Figure 4 biology-14-00136-f004:**
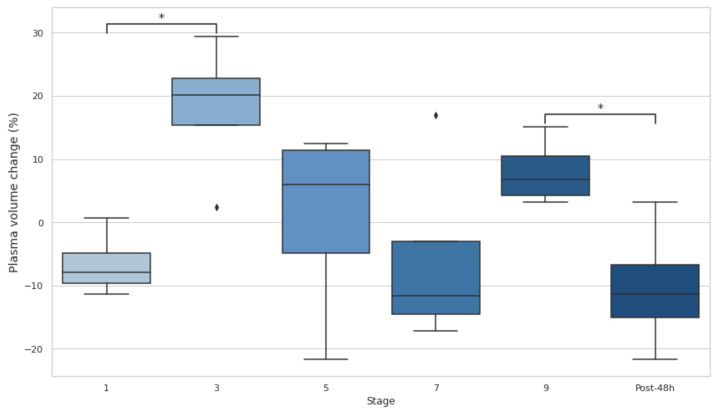
Plasma volume (%) at baseline, after stages 1, 3, 5, 7, and 9, and 48 h post-event. Post-48 h: 48 h post-event; * *p* < 0.05.

**Figure 5 biology-14-00136-f005:**
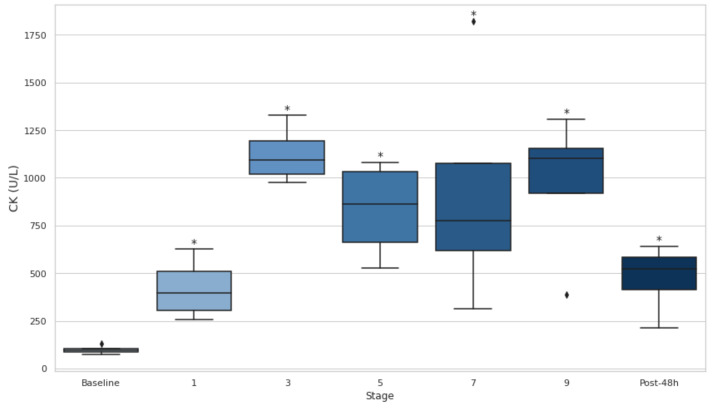
Serum creatine kinase at baseline, after stages 1, 3, 5, 7, and 9, and 48 h post-event. CK: creatine kinase. Post-48 h: 48 h post-event; * *p* < 0.05.

**Figure 6 biology-14-00136-f006:**
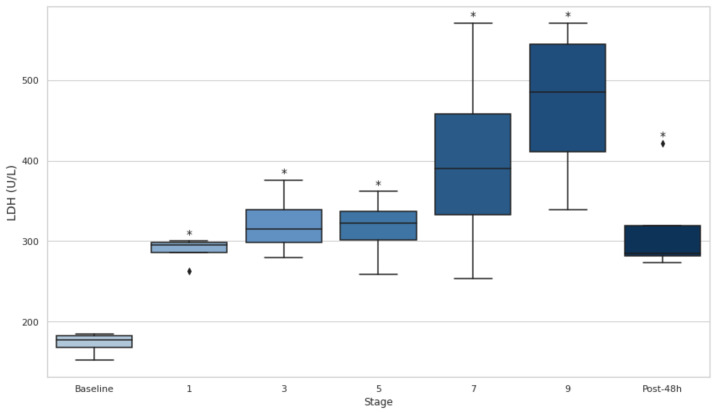
Serum lactate dehydrogenase at baseline, after stages 1, 3, 5, 7, and 9, and 48 h post-event. LDH: lactate dehydrogenase. Post-48 h: 48 h post-event; * *p* < 0.05.

**Figure 7 biology-14-00136-f007:**
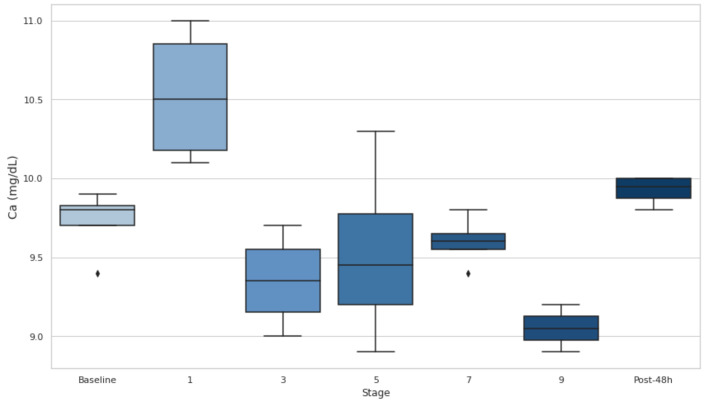
Serum calcium within the stages compared to the baseline. Ca: calcium. Post-48 h: 48 h post-event.

**Figure 8 biology-14-00136-f008:**
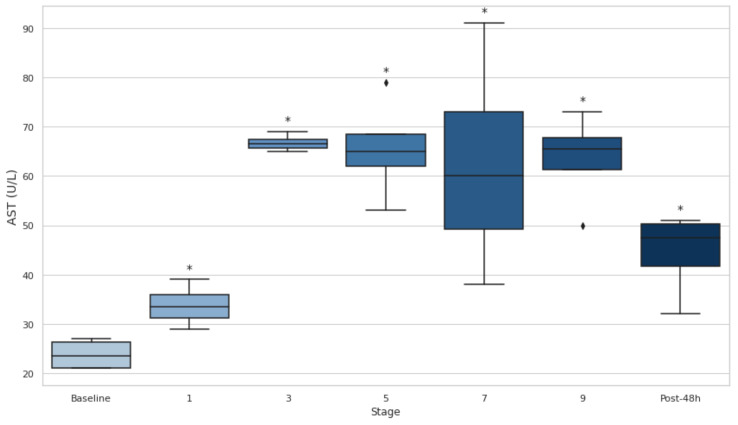
Serum aspartate aminotransferase at baseline, after stages 1, 3, 5, 7, 9 and 48 h post-event. AST: aspartate aminotransferase. Post-48 h: 48 h post-event; * *p* < 0.05.

**Figure 9 biology-14-00136-f009:**
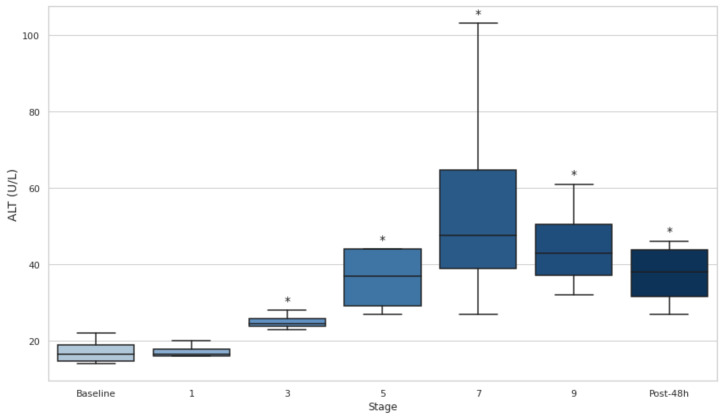
Serum alanine aminotransferase at baseline, after stages 1, 3, 5, 7, and 9, and 48 h post-event. ALT: alanine aminotransferase. Post-48 h: 48 h post-event; * *p* < 0.05.

**Figure 10 biology-14-00136-f010:**
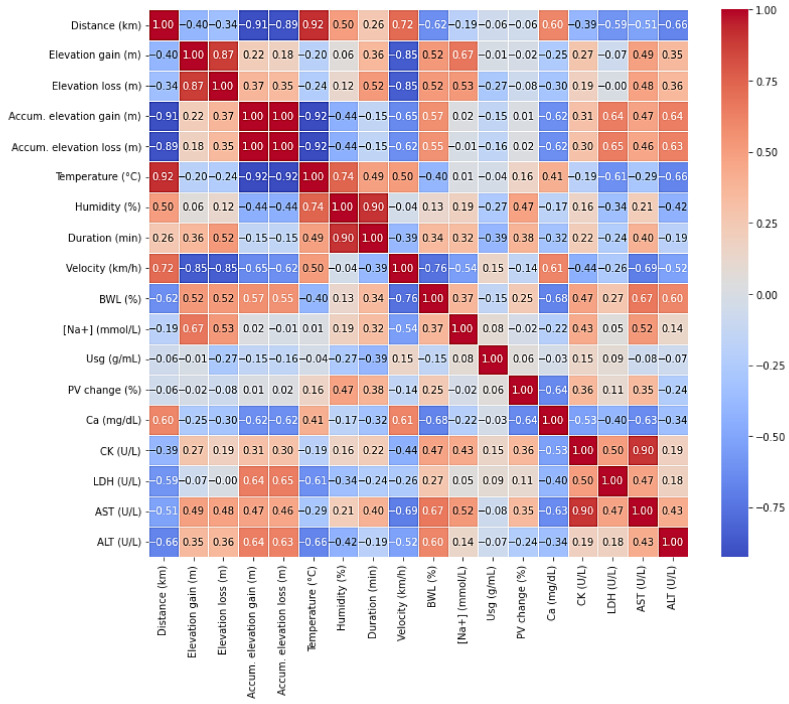
Heatmap of correlations between biomarkers and extrinsic event characteristics. BWL: body weight loss; Na: sodium; U_sg_: urine specific gravity; PV: plasma volume; Ca: calcium; CK: creatine kinase; LDH: lactate dehydrogenase; AST: aspartate aminotransferase; ALT: alanine aminotransferase; red range: positive correlation; blue range: negative correlation.

**Table 1 biology-14-00136-t001:** Stages’ characteristics.

Variable	Stage 1	Stage 3	Stage 5	Stage 7	Stage 9
Distance (km)	78.5	72.0	72.6	63.7	66.1
Elevation gain (m)	3136	4655	5411	5492	3361
Elevation loss (m)	3034	4044	6336	5163	3841
Temperature (°C)	26.0	26.0	21.7	9.0	11.2
Humidity (%)	58.0	62.0	61.0	55.0	58.0
Duration (hh:mm)	12:56	14:28	15:04	12:38	13:27
Velocity (km·h^−1^)	6.14	5.14	4.76	4.76	5.21

**Table 2 biology-14-00136-t002:** Participants’ demographic, anthropometric, and training characteristics.

Parameters	Mean ± SD
Age (years)	38 ± 4.11
Height (cm)	175.72 ± 3.65
Weight (kg)	70.09 ± 9.05
BMI	22.70 ± 2.05
Fat mass (%)	8.13 ± 0.68
Muscle mass (%)	46.75 ± 6.27
VO_2max_ (mL·kg^−1^·min^−1^)	61.17 ± 8.96
MAS (km·h^−1^)	16.91 ± 0.83
HR_max_ (beats·min^−1^)	187 ± 8.54
UT experience (years)	5 ± 1.26
Weekly volume (hours)	11.61 ± 2.22
Annual accumulated elevation gain (m)	116,615 ± 37,462

BMI: body mass index; VO_2max_: maximum oxygen consumption; MAS: maximal aerobic speed; HR_max_: maximum heart rate; UT: ultra-trail.

**Table 3 biology-14-00136-t003:** White blood cell lineage at baseline, after stages 1, 3, 5, 7, and 9, and 48 h post-event.

	Baseline	Stage 1	Stage 3	Stage 5	Stage 7	Stage 9	Post 48 h
WBC	5.73 ± 1.82	13.60 ± 1.42 *	10.25 ± 1.44 *	9.48 ± 1.16 *	11.28 ± 0.77 *	6.85 ± 1.23	7.70 ± 0.64 *
NEU	3.08 ± 0.66	10.20 ± 1.29	6.70 ± 0.86	6.28 ± 1.53	7.70 ± 1.85	3.93 ± 1.05	3.78 ± 0.17
LYM	2.58 ± 0.73	2.23 ± 0.93	7.80 ± 10.83	2.15 ± 0.44	2.48 ± 1.11	1.87 ± 0.66	3.03 ± 0.54
MON	0.50 ± 0.08	1.05 ± 0.31 *	0.95 ± 0.19 *	0.80 ± 0.22	0.78 ± 0.25	0.58 ± 0.17	0.53 ± 0.05
EOS	0.65 ± 0.77	0.05 ± 0.06 *	0.10 ± 0.08 *	0.53 ± 0.53	0.33 ± 0.13	0.38 ± 0.22	0.27 ± 0.13
BAS	0.03 ± 0.05	0.10 ± 0.00	0.08 ± 0.05	0.05 ± 0.06	0.08 ± 0.05	0.08 ± 0.05	0.08 ± 0.05
ERY	0.00 ± 0.00	0.00 ± 0.00	0.00 ± 0.00	0.00 ± 0.00	0.00 ± 0.00	0.00 ± 0.00	0.00 ± 0.00

WBC: white blood cells; NEU: neutrophils; LYM: lymphocytes; MON: monocytes; EOS: eosinophils; BAS: basophils; ERY: erythroblasts; * *p* < 0.05.

**Table 4 biology-14-00136-t004:** Red blood cell lineage at baseline, after stages 1, 3, 5, 7, and 9, and 48 h post-event.

	Baseline	Stage 1	Stage 3	Stage 5	Stage 7	Stage 9	Post 48 h
RBC	3.59 ± 2.39	4.98 ± 0.14	4.43 ± 0.25	4.56 ± 0.35	4.73 ± 0.06	4.44 ± 0.28	4.69 ± 0.20
Hb	14.60 ± 0.24	15.17 ± 0.39	13.77 ± 0.54	13.77 ± 1.01	14.20 ± 0.37	13.68 ± 0.51	14.67 ± 0.41
Hct	0.43 ± 0.01	0.44 ± 0.01	0.41 ± 0.01	0.41 ± 0.03	0.42 ± 0.02	0.41 ± 0.52	0.43 ± 0.02
MCV	91.25 ± 2.95	89.07 ± 3.04	91.55 ± 3.73	89.80 ± 0.45	89.82 ± 0.67	91.85 ± 2.55	92.88 ± 2.71
MCH	31.07 ± 1.02	30.53 ± 0.93	31.17 ± 1.53	30.18 ± 0.13 *	29.83 ± 0.41 *	30.73 ± 0.88	31.28 ± 0.88
MCHC	340.75 ± 1.71	342.75 ± 2.75	339.25 ± 6.13	336.50 ± 0.58 *	332.33 ± 2.05 *	335.50 ± 3.42	336.75 ± 1.71
RDW	13.10 ± 0.68	12.80 ± 0.66	13.20 ± 0.48	13.35 ± 0.70	13.27 ± 0.70	13.60 ± 0.39	13.27 ± 0.39
PLA	230.50 ± 47.57	233.75 ± 32.04	216.25 ± 35.67	204.75 ± 64.33	269.68 ± 40.78	266.25 ± 53.86	331.25 ± 47.79
MPV	8.63 ± 0.51	8.75 ± 0.42	8.67 ± 0.66	8.77 ± 0.43	8.63 ± 0.26	8.60 ± 0.64	8.20 ± 0.65

RBC: red blood cells; Hb: hemoglobin; Hct: hematocrit; MCV: mean corpuscular volume; MCH: mean corpuscular hemoglobin; MCHC: mean corpuscular hemoglobin concentration; RDW: red cell distribution width; PLA: platelets; MPV: mean platelet volume; * *p* < 0.05.

## Data Availability

The datasets generated during and/or analyzed during the current study are available from the corresponding author on reasonable request.

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
