# Peer review of "Impact of Ad Libitum Hydration on Muscle and Liver Damage and Electrolyte Balance in Ultra-Trail Events: A Heatmap Analysis of Biomarkers and Event Characteristics—A Pilot Study"

_biology, 2025, doi:10.3390/biology14020136_

Round 1
Reviewer 1 Report
Comments and Suggestions for Authors
The article was found to be both enjoyable and informative, with the methodological approach employed by Ultra-Trail, which incorporates advanced statistical calculations, greatly enhancing its appeal.
The following points may be considered:
1- The Monte Carlo method and the permutation should be accompanied by additional explanations. The reasons for using these methods should be mentioned briefly. Since these advanced statistical methods are rarely used in exercise physiology, readers may not be familiar with them. It is therefore necessary to provide additional explanations to facilitate communication with the method and the results of the article.
2- It is important to ascertain whether the amount of calories consumed during the given distance can be estimated. If this is indeed possible, the relevant data should be provided for each stage, with the necessary statistical work included if deemed appropriate. In addition, if data regarding daily calorie intake is available, this should also be incorporated. The incorporation of this additional information will serve to enhance the overall utility of the article.
3- Has statistical analysis been conducted in relation to the finishing time in question? In the event that heart rate measurements have been collected along the route, it would be beneficial to consider the calculation of the mean heart rate intensity for each stage of the race.
5- Utilizing the Dill & Costill formula, it is possible to measure the changes in plasma volume before and after each stage, thus providing valuable additional information. However, it is crucial to note that this approach is contingent upon the availability of pre- and post-stage data on the levels of hemoglobin and hematocrit. This approach would facilitate the incorporation of crucial variables such as cardiovascular risks associated with clot formation, platelet activity, and related coagulation indicators, thereby ensuring a comprehensive and nuanced assessment of the subject.
6- Has the amount of water consumed been measured despite ad libitum?
7- Recovery programs such as massage and cold therapy are common among ultra-trail runners. What about this research? have you used these types of recovery programs?
8- Has the quality of sleep and diet been checked during the recovery period?
9- In the absence of a control group, it is not possible to ascertain whether PDL would have resulted in an improvement or deterioration in the records. Consequently, it would be more appropriate to engage in a discussion on the matter.
11- Page 9, line 323, What do you mean by serum and surface in this section?
12- Line 30, the first sentence of the first paragraph and the sentence of the fifth paragraph (Line 86) are almost identical. It is recommended that this be corrected.
13- A brief discussion on the heat map would be beneficial, as it is presented in an effective manner.
14- In the discussion section, adaptations due to the high level of physical fitness of the subjects, which are likely to influence the obtained results, should be considered if possible.
Author Response
Dear reviewer 1,
Thank you for giving us the opportunity to submit a revised draft of our manuscript titled "Impact of Ad Libitum Hydration on Muscle, Liver Damage and Electrolyte Balance in Ultra-Trail: A Heatmap Analysis of Biomarkers and Event Characteristics. A Pilot Study” to the issue “New Insights into Skeletal Muscle Metabolism in Pathological and Physiological Conditions”.
We appreciate the time and effort that you have dedicated to providing your valuable feedback on our manuscript. Consequently, we have been able to incorporate changes to reflect most of the comments provided by you. We have highlighted the changes within the manuscript.
Here is a point-by-point response to your main notes and concerns. All these changes have been added to the main document and we have highlighted the corrections in yellow colour for you to find them easily. Additionally, some comments have been inserted. We hope you find it helpful.
1- The Monte Carlo method and the permutation should be accompanied by additional explanations. The reasons for using these methods should be mentioned briefly. Since these advanced statistical methods are rarely used in exercise physiology, readers may not be familiar with them. It is therefore necessary to provide additional explanations to facilitate communication with the method and the results of the article.
Dear reviewer, thank you for pointing this out. We agree with this comment and therefore we have added some extra lines to help clarify the statistics used (177-182)
2- It is important to ascertain whether the amount of calories consumed during the given distance can be estimated. If this is indeed possible, the relevant data should be provided for each stage, with the necessary statistical work included if deemed appropriate. In addition, if data regarding daily calorie intake is available, this should also be incorporated. The incorporation of this additional information will serve to enhance the overall utility of the article.
We see your point, but these data were not planned in the study´s goals. Consequently, we can add this information to the case study.
3- Has statistical analysis been conducted in relation to the finishing time in question? In the event that heart rate measurements have been collected along the route, it would be beneficial to consider the calculation of the mean heart rate intensity for each stage of the race.
We agree and we have added table 1. Stages´ characteristics. Time spent in each stage (duration) and velocity are included in the correlation’s heatmap. Unfortunately, Heart rate was not possible to calculate because of the lack of comfort for wearing such a device for a long race. And new devices such as wristbands were not available at that time.
5- Utilizing the Dill & Costill formula, it is possible to measure the changes in plasma volume before and after each stage, thus providing valuable additional information. However, it is crucial to note that this approach is contingent upon the availability of pre- and post-stage data on the levels of hemoglobin and hematocrit. This approach would facilitate the incorporation of crucial variables such as cardiovascular risks associated with clot formation, platelet activity, and related coagulation indicators, thereby ensuring a comprehensive and nuanced assessment of the subject.
Changes in plasma volume has been estimated in every stage and has been added to the heatmap. Plasma Volume has been estimated using this formula:
Corrected whole blood biomarkers – the equation of Dill and Costill revisited - Matomäki - 2018 - Physiological Reports - Wiley Online Library
We revised the new formula but we do not have the marker Blood biomarkers so we used the previous one
Corrected whole blood biomarkers – the equation of Dill and Costill revisited - Matomäki - 2018 - Physiological Reports - Wiley Online Library
6- Has the amount of water consumed been measured despite ad libitum?
We see your point, but we lack of this data we supposed that measuring the body weight in pre and post was enough as they were following an ad libitum strategy
7- Recovery programs such as massage and cold therapy are common among ultra-trail runners. What about this research? have you used these types of recovery programs?
We agree with your view, but we have not established any goals regarding this issue and all the runners used standardized cool down strategies.
8- Has the quality of sleep and diet been checked during the recovery period?
A brief questionnaire was given to the four runners. As they are very experienced and accustomed to sleeping outdoors regularly, their sleep patterns were completely normal; accordingly, no further examinations were carried out.
9- In the absence of a control group, it is not possible to ascertain whether PDL would have resulted in an improvement or deterioration in the records. Consequently, it would be more appropriate to engage in a discussion on the matter.
Que podemos añaddir enla discusion relacionado con el PDL ???
11- Page 9, line 323, What do you mean by serum and surface in this section?
Repeated word, paragraph has been rewritten
12- Line 30, the first sentence of the first paragraph and the sentence of the fifth paragraph (Line 86) are almost identical. It is recommended that this be corrected.
Totally agree line 86 has been removed
13- A brief discussion on the heat map would be beneficial, as it is presented in an effective manner.
We have rewritten the whole paragraph related to the results of the heat map (lines 318 – 331)
14- In the discussion section, adaptations due to the high level of physical fitness of the subjects, which are likely to influence the obtained results, should be considered if possible.
We have added in table 1 the runners´ experience in Ultra Trail. Añadir una línea de la experiencia de los corredores tabla 1 y de las adaptaciones crónicas al esfuerzo reflejadas en la literatura científica una referencia y listo
We hope you may find convenient the information added in this email, and please do not hesitate to contact us regarding any queries you might have.
Yours faithfully,

Reviewer 2 Report
Comments and Suggestions for Authors
Thank you for providing me the opportunity of reviewing the manuscript entitled "Impact of Ad Libitum Hydration on Muscle Damage and Electrolyte Balance in Ultra-Trail: A Heatmap Analysis of Biomarkers and Race Characteristics." The title is interesting. However, some points have been arised which may improve the content of manuscript.
Abstract
1. Statistical Reporting: The statistical significance (p-values) is reported different for findings (e,g, some findings have been missed or reported <0.05) and for some findings the exact amount of p values has been reported please make similar reporting findings by p value.
Introduction
2. Research Gap: While the introduction discusses the existing knowledge, it could strengthen the argument by explicitly stating the gap in the literature that this study aims to fill. A more direct statement of how this study contributes to the field would enhance the rationale for the research. It might be useful to briefly elaborate on the rationale behind the hypothesis, linking it back to the literature reviewed.
3. The novelty and necessity of this study must be included considering previous findings.
Methodological Details
4. Detailing the Race: While the description of the race (635 km with 40,586 m elevation gain/loss) is impressive, consider adding more context about the environmental conditions (e.g., temperature, humidity) during the race, as these factors can significantly impact hydration and performance.
5. Sample Size: The sample size of four participants is quite small. Consider discussing the limitations this may impose on the generalizability of the findings. You might also want to mention if there are any plans for future studies with larger sample sizes. Or publish the manuscript as case or pilot study.
6. Anthropometry and Body Weight Loss (BWL)Measurement Protocol: The protocol for measuring anthropometric data is clear, but it may be beneficial to specify the conditions under which these measurements were taken (e.g., time of day, fasting state) to ensure consistency.
7. Urine Analysis: The timing of urine sample collection after specific stages is clear. Consider discussing how hydration status can vary throughout the race and how this might affect the results.
Results
8. When mentioning correlation coefficients (e.g., r-values), providing context on what constitutes weak, moderate, or strong correlations could aid reader comprehension, especially for those less familiar with statistical analysis.
Discussion
9. Exercise-Associated Hyponatremia (EAH): The discussion on EAH is well-articulated. However, consider briefly explaining the physiological mechanisms behind EAH and its consequences on performance and health, as this could enhance reader understanding. You effectively compare your findings with previous studies. To strengthen this section, consider discussing potential reasons for the differences in EAH incidence among studies, such as variations in race conditions or participant characteristics.
10. BWL Results: The interpretation of BWL and its implications for hydration status is insightful. It might be helpful to discuss the practical implications of maintaining BWL below 2% in training and competition settings.
11. Mechanisms of Muscle Damage: While you mention the role of LDH and CK as markers of muscle damage, discussing the underlying physiological mechanisms that lead to these elevations (e.g., muscle fiber disruption, inflammation) could enhance the depth of this section.
12. The mention of calcium levels and their relationship with muscle function is important. Consider elaborating on the implications of hypocalcemia for performance and recovery.
13. Overall: strengthen discussion with related physiological mechanisms and practical recommendations
Author Response
Response to Reviewer 2 Comments
Dear Reviewer 2,
Thank you for giving us the opportunity to submit a revised draft of our manuscript titled "Impact of Ad Libitum Hydration on Muscle, Liver Damage and Electrolyte Balance in Ultra-Trail: A Heatmap Analysis of Biomarkers and Event Characteristics. A Pilot Study” to the issue “New Insights into Skeletal Muscle Metabolism in Pathological and Physiological Conditions”.
We appreciate the time and effort that you have dedicated to providing your valuable feedback on our manuscript. Consequently, we have been able to incorporate changes to reflect most of the comments provided by you. We have highlighted the changes within the manuscript.
Here is a point-by-point response to your main notes and concerns. All these changes have been added to the main document and we have highlighted the corrections in red colour for you to find them easily. Additionally, some comments have been inserted. We hope you find it helpful.
- Statistical Reporting: The statistical significance (p-values) is reported different for findings (e,g, some findings have been missed or reported <0.05) and for some findings the exact amount of p values has been reported please make similar reporting findings by p value.
We have added extra lines in the statistical analysis (l78 -184) detailing the procedures we carried out in detail according to your comments. We have used the same p-value in the whole article
Introduction
- Research Gap: While the introduction discusses the existing knowledge, it could strengthen the argument by explicitly stating the gap in the literature that this study aims to fill. A more direct statement of how this study contributes to the field would enhance the rationale for the research. It might be useful to briefly elaborate on the rationale behind the hypothesis, linking it back to the literature reviewed.
We have added a final sentence in the final paragraph of the introduction (lines 106 – 109)
- The novelty and necessity of this study must be included considering previous findings
We added lines 109-110 to underline the importance and the novelty of this approach by using a heatmap analysis
Methodological Details
- Detailing the Race: While the description of the race (635 km with 40,586 m elevation gain/loss) is impressive, consider adding more context about the environmental conditions (e.g., temperature, humidity) during the race, as these factors can significantly impact hydration and performance.
Fully agree, we added Table 1 where all the characteristics of the event are fully shown.
- Sample Size: The sample size of four participants is quite small. Consider discussing the limitations this may impose on the generalizability of the findings. You might also want to mention if there are any plans for future studies with larger sample sizes. Or publish the manuscript as case or pilot study.
- Statistics with Permutation tests and Monte Carlo Simulations for the small sample, to give more 'robustness' to the results. Explained in 'Statistics'
- I don't think we have put a section on 'Limitations'. Include it with the small sample size.
- Added in the title 'Pilot Study'.
- Anthropometry and Body Weight Loss (BWL)Measurement Protocol: The protocol for measuring anthropometric data is clear, but it may be beneficial to specify the conditions under which these measurements were taken (e.g., time of day, fasting state) to ensure consistency.
We added additional information regarding body weight measurement protocol (lines 143 – 148)
- Urine Analysis: The timing of urine sample collection after specific stages is clear. Consider discussing how hydration status can vary throughout the race and how this might affect the results.
We fully agree however due to the length of the event and the resources available only pre and post-measurements were possible. We would like to study urine analysis during the event and we would consider these results in future research.
Results
- When mentioning correlation coefficients (e.g., r-values), providing context on what constitutes weak, moderate, or strong correlations could aid reader comprehension, especially for those less familiar with statistical analysis.
Lines 188 -189 have been added to explain Cohen´s values as small, medium and large.
Discussion
- Exercise-Associated Hyponatremia (EAH): The discussion on EAH is well-articulated. However, consider briefly explaining the physiological mechanisms behind EAH and its consequences on performance and health, as this could enhance reader understanding. You effectively compare your findings with previous studies. To strengthen this section, consider discussing potential reasons for the differences in EAH incidence among studies, such as variations in race conditions or participant characteristics.
We see your point, however, the limitation in the maximum words of the article prevent us from adding more information and we decided to focus on the results found in the heatmap.
- BWL Results: The interpretation of BWL and its implications for hydration status is insightful. It might be helpful to discuss the practical implications of maintaining BWL below 2% in training and competition settings.
We mention the need to use both hydration strategies to prevent from suffering EAH and ER and we underline the importance of &BDWL as the gold standard to prevent runners form suffering EAH
- Mechanisms of Muscle Damage: While you mention the role of LDH and CK as markers of muscle damage, discussing the underlying physiological mechanisms that lead to these elevations (e.g., muscle fibre disruption, inflammation) could enhance the depth of this section.
We added lines 437 and 438 to add some practical use of these findings
- The mention of calcium levels and their relationship with muscle function is important. Consider elaborating on the implications of hypocalcemia for performance and recovery.
We show the result but we cannot deep in this result because of the objectives of the study
- Overall: strengthen discussion with related physiological mechanisms and practical recommendations
We have added new practical recommendations and offer new physiological mechanisms
We hope you may find convenient the information added in this email, and please do not hesitate to contact us regarding any queries you might have.
Yours faithfully,
Reviewer 3 Report
Comments and Suggestions for Authors
The authors have tried to study the effect if ultra trail running on athletes and found that Ultra-trail races (UT) present risks of dehydration (DH), overhydration (OH), exercise-associated hyponatremia (EAH), and exertional rhabdomyolysis (ER). This agrees with previous studies. However, it would have been useful to do a recovery analysis to understand the persistence of those temporary effects.
They studied four trained male athletes during a nine-stage UT (635 km, 40,586 m elevation gain) found no cases of EAH or hypernatremia, but dehydration exceeded thresholds in early stages, and muscle damage markers (CK, LDH) significantly increased throughout, persisting post-event. Elevation changes worsened fluid loss and muscle injury.
Specific comments:
1. Four athletes is considered a low sample size to come to conclude anything concretely and it'll be good to see similar trends hold up for other athletes as well.
2. It would be useful to have a more comprehensive outlook on biomarkers, especially those pertaining to liver and heart.
3. Seeing an overall picture of blood including RBCs and WBCs will be important, including changes during race and recovery, like changes in Hb, RBC count, Hct, mean cell volume, WBC counts for Neutrophils, Lymphocytes etc for obtaining a more comprehensive understanding.
4. There is no mention or measurement of metabolic changes during the race and recovery. It's important to understand the lipid profile changes, liver enzyme changes during the race and whether they change or increase and plateau and how long they take to return to baseline. This can help inform the personalized hydration plan too.
5. The authors found that Ad libitum hydration reduced EAH risk but was insufficient to prevent DH or muscle damage, highlighting the need for personalized hydration and recovery strategies. However, there's no control on water consumption per kg bodyweight or any comparison as such to make these claims. Ad libitum hydration is very subjective and while personalized hydration makes sense intuitively, the authors haven't performed any experiment to justify the statement.
Comments on the Quality of English LanguageIt will benefit from proofreading and editing to ensure that the novelty of the study comes across.
Author Response
Dear reviewer 3,
Thank you for giving us the opportunity to submit a revised draft of our manuscript titled "Impact of Ad Libitum Hydration on Muscle, Liver Damage and Electrolyte Balance in Ultra-Trail: A Heatmap Analysis of Biomarkers and Event Characteristics. A Pilot Study”.
We appreciate the time and effort that you have dedicated to providing your valuable feedback on our manuscript. Consequently, we have been able to incorporate changes to reflect most of the comments provided by you. We have highlighted the changes within the manuscript.
Here is a point-by-point response to your main notes and concerns. All these changes have been added to the main document and we have highlighted the corrections in red colour for you to find them easily. Additionally, some comments have been inserted. We hope you find it helpful.
Right now, we would like to offer a detailed explanation of every one of your corrections:
Specific comments:
- Four athletes is considered a low sample size to come to conclude anything concretely and it'll be good to see similar trends hold up for other athletes as well.
We see your point, but we lack of this data we supposed that measuring the body weight in pre and post was enough as they were following an ad libitum strategy
- It would be useful to have a more comprehensive outlook on biomarkers, especially those pertaining to the liver and heart.
We have added liver biomarkers in our study and even in the title. We fully support your idea because liver damage is still unclear in these races and it deserves further investigation. Heart damage biomarkers were not established as an objective of this study so we do not have data regarding this kind of damage
- Seeing an overall picture of blood including RBCs and WBCs will be important, including changes during race and recovery, like changes in Hb, RBC count, Hct, mean cell volume, WBC counts for Neutrophils, Lymphocytes etc for obtaining a more comprehensive understanding.
We agree and we have added Table 4 including RBCs and WBCs
- There is no mention or measurement of metabolic changes during the race and recovery. It's important to understand the lipid profile changes, liver enzyme changes during the race and whether they change or increase and plateau and how long they take to return to baseline. This can help inform the personalized hydration plan too.
We would have liked to add parameters regarding to metabolic substrates or liver enzymes however because of the length and duration of the test was not possible. Maybe in a laboratory test could be implemented.
- The authors found that Ad libitum hydration reduced EAH risk but was insufficient to prevent DH or muscle damage, highlighting the need for personalized hydration and recovery strategies. However, there's no control on water consumption per kg bodyweight or any comparison to make these claims. Ad libitum hydration is very subjective and while personalized hydration makes sense intuitively, the authors haven't performed any experiment to justify the statement.
We fully support your idea but the duration and the length of the event prevented us from registering all the beverages consumed. We followed the ad libitum protocol and trusted in the runners' great experience. Had they been amateur we would have registered exhaustively the quantity of beverages drunk
We hope you may find convenient the information added in this email, and please do not hesitate to contact us regarding any queries you might have.
Yours faithfully,
Round 2
Reviewer 2 Report
Comments and Suggestions for Authors
Thank you for considering revisions
still you need to consider: 1-regarding the caption of figure 1 the last sentence with related reference seems not necessary. it can be moved to the text. Also I can not see the interpratation of small , medium or large correlation in the figure or text. 2- regading discussion: This manuscript will be published in biology journal, so justifying findings with biological mechanisms seems important. Pearson correlation coefficients were interpreted as small (r=0.10), medium (r=
352
0.30), and large (r=0.50).
Author Response
Question 1
Still you need to consider:
1-regarding the caption of figure 1 the last sentence with related reference seems not necessary. it can be moved to the text. Also I can not see the interpratation of small , medium or large correlation in the figure or text.
1. The last sentence in the caption of Figure 1 has been removed and incorporated into the ‘Statistical Analysis’ section of the text. Specifically, we now state: ’Effect sizes were calculated using Cohen’s D, where values of 0.2-0.49, 0.5-0.79, and 0.8-1.0 were interpreted as small, medium, and large effects, respectively. Additionally, Pearson correlation coefficients were interpreted as small (r = 0.10-0.29), medium (r = 0.30-0.49), and large (r ≥ 0.50) [29].’
2- Regarding discussion: This manuscript will be published in biology journal, so justifying findings with biological mechanisms seems important.
We hope you may find convenient the information added in this email, and please do not hesitate to contact us regarding any queries you might have.
Yours faithfully,
Miguel Lecina
University of Zaragoza

Reviewer 3 Report
Comments and Suggestions for Authors
Thank you for addressing my comments and including more blood cell data. It adds to the appeal of the manuscript.
Author Response
Dear Reviewer 3
Thank you for your help in improving our article. As a result of the addition of the blood cells, the article seems much more accurate and scientific.
Yours faithfully,
Miguel Lecina
University of Zaragoza